# Understanding the Continuum between High-Risk Myelodysplastic Syndrome and Acute Myeloid Leukemia

**DOI:** 10.3390/ijms24055018

**Published:** 2023-03-06

**Authors:** Phaedon D. Zavras, Ilias Sinanidis, Panagiotis Tsakiroglou, Theodoros Karantanos

**Affiliations:** Division of Hematologic Malignancies and Bone Marrow Transplantation, Sidney Kimmel Comprehensive Cancer Center, Johns Hopkins University, Baltimore, MD 21231, USA

**Keywords:** MDS, AML with MDS-related changes, molecular alterations, classification, therapeutic approaches

## Abstract

Myelodysplastic syndrome (MDS) is a clonal hematopoietic neoplasm characterized by bone marrow dysplasia, failure of hematopoiesis and variable risk of progression to acute myeloid leukemia (AML). Recent large-scale studies have demonstrated that distinct molecular abnormalities detected at earlier stages of MDS alter disease biology and predict progression to AML. Consistently, various studies analyzing these diseases at the single-cell level have identified specific patterns of progression strongly associated with genomic alterations. These pre-clinical results have solidified the conclusion that high-risk MDS and AML arising from MDS or AML with MDS-related changes (AML-MRC) represent a continuum of the same disease. AML-MRC is distinguished from de novo AML by the presence of certain chromosomal abnormalities, such as deletion of 5q, 7/7q, 20q and complex karyotype and somatic mutations, which are also present in MDS and carry crucial prognostic implications. Recent changes in the classification and prognostication of MDS and AML by the International Consensus Classification (ICC) and the World Health Organization (WHO) reflect these advances. Finally, a better understanding of the biology of high-risk MDS and the mechanisms of disease progression have led to the introduction of novel therapeutic approaches, such as the addition of venetoclax to hypomethylating agents and, more recently, triplet therapies and agents targeting specific mutations, including FLT3 and IDH1/2. In this review, we analyze the pre-clinical data supporting that high-risk MDS and AML-MRC share the same genetic abnormalities and represent a continuum, describe the recent changes in the classification of these neoplasms and summarize the advances in the management of patients with these neoplasms.

## 1. Introduction

Myelodysplastic syndrome (MDS) is a heterogeneous group of clonal hematopoietic stem cell disorders characterized by bone marrow dysplasia, hematopoiesis failure and high risk of progression to acute myeloid leukemia (AML) [1,2,3]. One out of three patients diagnosed with MDS will progress to AML, characterized by an increased percentage of blasts over 20% [4]. Based on the 2016 world classification of hematologic malignancies, AML arising from MDS was classified as a distinct clinicopathologic entity entitled “AML with MDS-related changes” (AML-MRC) [5]. AML-MRC is associated with an overall poor response to both induction chemotherapy and low-intensity therapy, a high incidence of relapse and worse overall survival compared to patients with de novo AML [6]. Due to the poor outcomes of individuals with this disease, it has been the focus of intense research at both the pre-clinical and clinical levels.

Recent large-scale studies have demonstrated that various cytogenetic abnormalities and gene mutations, such as deletion of 5q or 7q/7, mutations in spliceosome genes and genes encoding epigenetic modifiers, are common in MDS and AML-MRC but appear to be less frequent in de novo AML [7,8,9]. On the contrary, other genomic alterations, such as mutations in transcriptional factors or genes encoding proteins involved in signal transduction, are identified more frequently in AML-MRC compared to MDS [8,10,11], suggesting that these are probably later biological events during disease progression. Single-cell multi-omic studies have recently revealed specific patterns of clonal evolution driving the progression of MDS to AML-MRC [12,13], further highlighting that high-risk MDS and AML-MRC probably represent different stages of the same myeloid disease.

These research advances have significantly improved our understanding of the biology of high-risk MDS and its transformation to AML-MRC, leading to significant improvements in the classification of these presentations and a more comprehensive therapeutic approach incorporating novel agents and broadening opportunities for clinical trials. These changes are reflected in the recently updated classification of myeloid neoplasms by the International Consensus Classification (ICC) and the World Health Organization (WHO) [14,15]. Notably, the recognition of a continuum between high-risk MDS and AML-MRC has yielded the enrollment of high-risk MDS patients into AML clinical trials and the enrollment of AML-MRC patients into MDS clinical trials, which is expected to tremendously improve our understanding of this disease’s biology and the survival of these patients.

In this review, we summarize the existing pre-clinical studies that support the notion that high-risk MDS and AML-MRC represent a continuum and provide a novel understanding of the mechanisms of disease progression. Moreover, we describe the most recent changes in the classification of MDS and AML-MRC, as well as the implications of these advances in the management of patients with these myeloid neoplasms.

## 2. Genetic Alterations Underlie MDS Progression to AML-MRC

### 2.1. Chromosomal Abnormalities

Chromosomal abnormalities and copy number alterations are common in MDS and AML-MRC and have important prognostic implications. MDS and AML-MRC share various chromosomal abnormalities, such as deletion of 5q, 7/7q, 20q and complex karyotype, that typically lead to copy number alterations as opposed to de novo AML, which is frequently characterized by balanced re-arrangements, such as translocations 15;17 and 8;21 and inversion 16 [7]. Most of the chromosomal abnormalities identified in patients with MDS have critical prognostic implications and are associated with variable risk of disease progression.

Deletion of 5q is the most common chromosomal abnormality in MDS, particularly among patients with lower-risk disease, and has been associated with an overall good prognosis [16,17,18]. Hematopoietic cells with 5q deletion have defective ribosomal biogenesis, resulting in elevated levels of free ribosomal proteins in their cytoplasm, which bind and promote the degradation of MDM2, a key P53 regulator [19]. This leads to P53 activation, which promotes cell-cycle arrest and induces apoptosis of erythroid progenitors and ineffective erythropoiesis [20]. These molecular events render P53 activity critical for the prevention of this disease’s progression to AML. Indeed, analysis of 55 individuals with low or intermediate-1 MDS with 5q deletion showed that *TP53* mutation is associated with an increased risk of AML transformation [21]. Interestingly, *TP53* mutations were identified in the majority of patients at an early disease stage and were always detectable before AML transformation [21]. Thus, the presence of a *TP53* mutation should be an alarm for a high risk of progression in MDS with 5q deletion, even if identified at an earlier disease stage. It should be also noted that MDS patients with a complex karyotype, including 5q deletion, have a poor response to chemotherapy or lenalidomide and a dismal prognosis, which are linked to genomic instability due to P53 dysfunction [22]. Thus, MDS with 5q deletion in the context of a complex karyotype should not be managed as a low-risk disease; it needs to be approached as a high-risk disease. Consistently, 5q deletion is associated with poor survival, high genomic complexity and biallelic *TP53* variants among patients with high-risk MDS and AML-MRC [23].

Loss of chromosome 7 or 7q is the second most common chromosomal abnormality among MDS patients. It is associated with overall poor prognosis and a higher risk of progression to AML-MRC with worse outcomes for patients with loss of the entire chromosome 7 [24]. Similarly to 5q, the loss of 7q is associated with high genomic complexity and particularly short survival among patients with AML-MRC [23]. A number of genes with a critical role in the pathogenesis of myeloid malignancies, such as *SAMD9*, *SAMD9L*, *EZH2*, *CUX1* and *MLL3*, are located on chromosome 7 [25]. Most of these genes encode proteins that act as tumor suppressors, and deletion or missense mutations provide survival and growth benefits to hematopoietic stem and progenitor cells [26,27]. Recent data also support the notion that loss of chromosome 7 or 7q is associated with significant alterations in the expression of interferon-gamma pathway genes, promoting an immunosuppressive microenvironment enriched in T regulatory immune cells, potentially conferring with the inferior outcomes of patients with myeloid neoplasms with these abnormalities [28]. Overall, these results support that loss of chromosome 7 or 7q contributes to the pathogenesis and, more importantly, to the progression of MDS to AML-MRC.

The presence of three or more chromosomal abnormalities is defined as a complex karyotype (CK) and is always considered a predictor of adverse outcomes, independent of actual specific abnormalities [29]. CK, particularly monosomal karyotype (MK), is strongly associated with one or more mutations in *TP53* and frequently with loss of heterozygosity (LOH) of this gene [30,31,32]. A number of studies have highlighted that CK is associated with particularly poor survival and a high risk of transformation to AML among MDS patients, independently of blast percentage and other clinical features such as cytopenias and transfusion dependence [33,34,35]. These findings further support that the notion that the biology of MDS and its progression is defined by genetic alterations, probably at the earlier stages of this disease.

### 2.2. Somatic Mutations

Next-generation sequencing studies have demonstrated that MDS and AML-MRC share mutations in genes implicated in cellular processes, such as RNA splicing, epigenetic and transcriptional regulation, and DNA damage repair [8]. Based on an analysis of 299 AML patients, mutations in spliceosome genes such as *SRSF2*, *SF3B1*, *ZRSF2* and *U2AF1*, *EZH2*, *BCOR* and *STAG2* are >95% specific for AML-MRC and can distinguish AML-MRC from de novo AML, even in the absence of an antecedent MDS diagnosis [8]. The identification of mutations in genes implicated in different cellular processes highlights the progression of clonal myeloid neoplasms from earlier stages, such as clonal hematopoiesis of indeterminate potential (CHIP) to MDS and AML-MRC. Particularly, mutations in epigenetic modifiers, such as *TET2* and *DNMT3A* and *TP53*, are common across a spectrum of clonal hematopoietic disorders, such as CHIP, clonal cytopenias, MDS and AML-MRC [36]. On the contrary, mutations in spliceosome genes are more common in MDS and AML-MRC [37,38], while mutations in transcriptional factors, such as *RUNX1* and *CEBPA*, and activating signaling genes, such as *NRAS* and *FLT3*, are more common in AML-MRC [8,9]. These findings suggest that clonal evolution through the acquisition of additional somatic mutations probably drives the progression of these clonal myeloid neoplasms.

*SF3B1* is commonly mutated in MDS [39], with the majority of mutations being missense substitutions affecting the spliceosome machinery, resulting in altered proteome [40]. *SF3B1* mutations have been associated with MDS with ring sideroblasts and a relatively low risk of AML transformation [41]. However, specific substitutions, such as K666N hotspot mutation, result in distinct patterns of RNA splicing and increased risk of progression to AML-MRC [42]. *SRSF2* mutations are found in 10–15% of patients with MDS and are associated with older age, higher levels of hemoglobin and a normal karyotype [43]. *SRSF2* mutations frequently co-occur with *RUNX1*, *IDH2* and *ASXL1* mutations and are associated with a higher risk of transformation to AML-MRC [43,44]. The adverse biology of MDS with *SRSF2* mutations could be associated with the altered splicing of genes such as *EZH2* [45] and their implications in DNA damage accumulation through alterations of the P53 function [46]. *U2AF1* mutations occur in about 5–10% of MDS patients and are associated with a high risk of disease progression to AML-MRC [47]. Aberrant splicing due to *U2AF1* mutations results in suppression of ATG7 levels and reduced autophagy, causing increased oxidative stress and chromosomal instability [48]. A recent study also revealed that *U2AF1* mutations cause an increase in the expression of the long isoform of interleukin-1 receptor-associated kinase 4 (IRAK4), which promotes the activation of nuclear factor kappa-light-chain-enhancer of B cells (NF-kB) that have been implicated in leukemic growth [49]. *ZRSR2* is mutated in about 5% of patients with MDS, affect predominantly men and cause aberrant splicing by retaining U12-dependent introns [31,32,33,34,35,36,37,38,39,40,41,42,43,44,45,46,47,48,49,50]. While most studies indicate a negative impact of *ZRSR2* mutations on the survival of MDS patients, the impact is not as strong compared to *SRSF2* and *U2AF1* mutations [51]. These results highlight that mutations in spliceosome genes can mediate leukemic growth and promote the progression of MDS to AML-MRC.

*STAG2* encodes a subunit of the cohesin complex, which regulates the separation of sister chromatids during mitosis [52] and is mutated in close to 10% of patients with MDS [53] with male predominance [54]. *STAG2* mutations have been associated with poor overall survival among MDS patients [53] but have not been directly linked to progression to AML-MRC. It has been hypothesized that the haploinsufficiency of genes encoding subunits of the cohesin complex affects the expression of genes that are essential for lineage priming and differentiation [55]. Thus, mutations in these genes likely cause abnormal hematopoietic stem cell maturation contributing to dysplasia, but cooperating oncogenic mutations are required for progression to AML-MRC [55]. Indeed, *RUNX1*, *CEBPA*, *NPM1*, and *RAS* mutations co-occur with *STAG2* mutations in AML [9,56,57].

EZH2 is a core component of polycomb group complex 2 (PRC2), which is responsible for trimethylation of lysine 27 of histone 3 and plays a critical role in epigenetic gene silencing [58]. *EZH2* is mutated in 4–5% of MDS patients [59] and is frequently co-mutated with *ASXL1* and *RUNX1* [60,61]. Concurrent *TET2* and *EZH2* deletion is sufficient to induce an MDS/myeloproliferative phenotype in mice [62], and in a *RUNX1* mutant model, *EZH2* loss promotes myelodysplasia development but inhibits transformation to AML [63]. In fact, EZH2 is upregulated in AML with CK [64] and EZH2 deletion suppresses the proliferation of MLL-AF9-transformed cells and delays AML transformation in transgenic mice [65]. Consistently, *EZH2* mutations have been associated with poor overall survival in patients with chronic myeloid neoplasms [30,54,66] and poor response to hypomethylating agents [67] but not with a higher risk of AML-MRC transformation. Thus, similarly to *STAG2* mutations, it can be hypothesized that *EZH2* mutations are early events in MDS [68], and additional oncogenic mutations are required for transformation to AML-MRC.

The *BCOR* gene is located on the chromosome X and encodes transcription factor that is essential for normal embryonic development [69]. *BCOR* mutations are detected in less than 5% of patients with MDS and are typically frameshift insertions or deletions, or stopgain or non-sense mutations, causing a suppression of function of the BCOR protein in hematopoietic cells [70,71]. Recent studies have demonstrated that loss of BCOR function is associated with altered activity of polycomb group repressive complex 1 (PRC1), resulting in induced myeloid cells’ proliferation [71]. Most of the studies on large cohorts have demonstrated that BCOR mutations have a neutral impact on overall survival of MDS patients [72,73].

*RUNX1* encodes the alpha subunit of the core binding transcription factor and is implicated in the differentiation of hematopoietic stem cells [74]. Germline point mutations of *RUNX1* have been identified in an autosomal dominant platelet disorder with a high risk of transformation to AML [75]. *RUNX1* mutations occur in 10–15% of MDS and have been associated with thrombocytopenia, poor survival with a high risk of progression to AML-MRC, co-occurrence of RAS mutations and loss of chromosome 7/7q [76,77,78]. Recent data have also associated *RUNX1* mutations with poor response to hypomethylating agents [67,68,69,70,71,72,73,74,75,76,77,78,79]. Recently, it was reported that *RUNX1* mutation is associated with rapid progression of low-risk MDS and that mutations in this gene in CD34+ cells from low-risk MDS patients result in dysregulated DNA damage repair and cellular senescence [78]. The authors also found that the transcriptional profiles of CD34+ cells from low-risk MDS patients with *RUNX1* mutation resemble those of high-risk MDS at diagnosis [78]. These findings further suggest that specific genomic alterations identified even at the earlier stage of MDS can alter disease biology and induce disease rapid progression.

*NRAS* and *KRAS* mutations occur in about 4–5% of patients with MDS and are associated with higher white blood cell counts and a higher risk of transformation to AML-MRC [80,81,82,83]. Mutations in *RAS* genes result in RAS/Raf/MEK and PI3K/AKT signaling activation and are sufficient to induce a myeloproliferative phenotype or even AML in mice [84,85]. Similarly, *FLT3-ITD* mutations are relatively uncommon among MDS patients with a frequency of 0.6–6% [86] and have been associated with higher blasts percentage, increased risk of leukemic transformation and worse overall survival [87]. FLT3-ITD mutations in hematopoietic stem and progenitor cells cause constitutive autophosphorylation of the FLT3 receptor, leading to the activation of downstream signaling, such as STAT3/5, MAPK and PI3K/AKT [88]. Analysis of 278 MDS patients with low- or intermediate-1-risk MDS revealed that the detection of a *RAS* or *FLT3-ITD* mutation is associated with a particularly high risk of progression to AML-MRC with very poor response to treatment with hypomethylating agents [11]. Similarly, the detection of *RAS* or F*LT3-ITD* mutations at the time of transformation of MDS to AML-MRC is correlated with significantly worse survival [89]. Thus, independently of the disease stage at diagnosis, the acquisition of these signaling genes mutations significantly alters disease biology and is connected with poor outcomes.

*TP53* is a tumor-suppressor gene located on the short arm of chromosome 17 and encodes transcription factor P53, which is critical for cell-cycle arrest, DNA repair and apoptosis in the setting of DNA damage [90]. As a result, cancers with *TP53* mutation accumulate DNA damage and demonstrate a poor response to cytotoxic therapy [91]. *TP53* alterations are discovered in about 5–10% of MDS patients [34,35,36,37,38,39,40,41,42,43,44,45,46,47,48,49,50,51,52,53,54,55,56,57,58,59,60,61,62,63,64,65,66,67,68,69,70,71,72,73,74,75,76,77,78,79,80,81,82,83,84,85,86,87,88,89,90,91,92], with the majority of cases having “multi-hit” involvement with more than one genomic and/or chromosome 17 abnormalities and fewer alterations in other genes [92]. A recent analysis of 3324 MDS patients revealed that high-risk presentation with CK, high incidence of transformation to AML-MRC and poor overall survival are associated with multi-hit *TP53* and not monoallelic *TP53* alterations [92]. It should be noted that these associations were independent of the Revised International Prognostic Scoring System (IPSS-R) [92], once more indicating that, independent of the disease stage at diagnosis, the molecular alterations that affect malignant cell biology define disease course and outcomes.

The incidence, biological mechanisms and clinical significance of genetic abnormalities in MDS and AML-MRC are summarized in Table 1.

### 2.3. Single-Cell Analysis Reveals Biologic Mechanisms Implicated in MDS Progression to AML-MRC

Single-cell DNA and RNA sequencing has significantly improved our understanding of heterogeneity and clonal evolution in myeloid neoplasms [94,95,96]. Chen et al. utilized longitudinal, paired samples from seven MDS patients who progressed to AML-MRC and performed ensemble deep sequencing and single-cell sequencing of sorted pre-malignant and malignant stem cells based on the expression of previously identified stem cell markers and blasts [95]. The authors demonstrated a significantly higher complexity of sub-clonal mutations in stem cells at the MDS stage compared to blast cells [95]. Moreover, the authors presented data that support a model of parallel clonal evolution at the stem cell level during MDS progression [95]. Particularly, all the patients studied demonstrated a highly diverse pool of pre-malignant MDS stem cells with most of the patients showing relatively early branching of these pre-malignant MDS stem cells towards progression to AML stem cells [95]. These findings suggest that the drivers of progression are probably present in MDS stem cells even at the earliest stages and dictate the natural history of this disease. Finally, the identification of actionable mutations in MDS stem cells could provide a particularly promising opportunity for the early prevention of disease progression to AML-MRC.

Guess et al. performed single-cell DNA sequencing of paired MDS and AML-MRC to better characterize clonal shifts upon MDS progression [12]. The authors found two different patterns of clonal evolution during MDS progression to AML-MRC [12]. The first pattern, called the static group, was characterized by relatively stable clonal architecture during progression [12]. In this group, patients had founder mutations in DNA methylation genes, such as *DNMT3A*, *TET2* and *IDH1/2*, and potentially epigenetic alterations drive progression to AML-MRC through blast growth [12]. On the contrary, the second pattern, called the dynamic group, was characterized by significant clonal architectural changes defined by new chromosomal abnormalities and TP53 mutations or genomic alterations with more prominent mutations in signaling genes [12]. Single-cell transcriptomic analysis demonstrated that pathways such as cell-cycle regulation via E2F activation and inflammatory signaling mediated by cytokine receptors are amongst the top upregulated cellular pathways associated with transformation to AML-MRC [12]. These results are consistent with mechanistic studies highlighting that inflammatory signaling is linked with cell-cycle progression promoting the progression of MDS to AML-MRC and affecting the disease biology and sensitivity to current therapies [93,97,98,99,100,101].

## 3. Changes in the Definition and Classification of MDS and AML-MRC

The International Consensus Classification (ICC) and the World Health Organization (WHO) have recently updated the classification of myeloid neoplasms that come with a better understanding of the biology of the disease and therapeutic advancements. Compared to the WHO 2016 classification, the ICC introduced changes in the predefining blast cut-offs, and both committees incorporated genetic alterations of the diseases.

### 3.1. MDS and AML with Myelodysplasia-Related Gene Mutations as a Disease Continuum

The new guidelines recognize clonal hematopoiesis (CH) as the precursor of cytopenias, ranging from clonal cytopenia of undetermined significance (CCUS) to MDS. Cases with cytopenia but without dysplasia lack three cytogenetic abnormalities, namely del(5q), –7/del(7q) and complex karyotype, and are now classified under CCUS. The definitions of the pre-malignant clonal hematopoiesis disease spectrum are summarized in Table 2. The ICC and the WHO recognize MDS as myeloid neoplasms characterized by clonal hematopoiesis, dysplastic changes in the BM of PB and cytopenia in at least one cell line [15,16,17,18,19,20,21,22,23,24,25,26,27,28,29,30,31,32,33,34,35,36,37,38,39,40,41,42,43,44,45,46,47,48,49,50,51,52,53,54,55,56,57,58,59,60,61,62,63,64,65,66,67,68,69,70,71,72,73,74,75,76,77,78,79,80,81,82,83,84,85,86,87,88,89,90,91,92,93,94,95,96,97,98,99,100,101,102]. Clonality is established via next-generation sequencing (NGS) or karyotype. In cases where clonality cannot be proven, the diagnosis of MDS can still be made with qualifying dysplasia and cytopenia(s). On the other hand, irrespective of the presence of dysplasia, MDS-defining genetic abnormalities are sufficient for diagnosis, especially in the setting of persistent cytopenia.

Emphasis has been placed on the molecular categorization of MDS and AML in the new ICC guidelines because the genetic footprint seems to be a stronger predictor of prognosis than the degree of dysplasia [103].

The previously defined categories of MDS with excess blasts (EBs) have been refined. Recent data show that MDS with a higher blast percentage is more likely to transform to AML and carries a worse prognosis, similar to overt AML [104,105]. To more accurately reflect the continuum between MDS and AML, a new entity “MDS/AML” has been suggested to define MDS with blasts between 10 and 19%, lacking AML-defining genetics. Patients with MDS/AML should be eligible for either MDS or AML treatment modalities or clinical trials. MDS/AML is further subclassified into cases with mutated TP53, cases with myelodysplasia-related gene mutations, myelodysplasia-related cytogenetic abnormalities and MDS/AML-NOS. Blasts of greater than 20% of define AML in these cases. AML positive for the Philadelphia chromosome (Ph+) or *BCR:ABL1*/t(9;22)(q34.1;q11.2) is an exception and is still defined as blasts ≥20%. The MDS/AML category is not applicable in this case so as to distinguish it from the progression of Ph+ CML. For cases with 5–9% and 2–9% blasts in BM and PB, respectively, the term MDS with EB is used.

The previously defined category of AML-MRC has been eliminated by the ICC. Genetic characteristics, rather than clinical history (i.e., de novo AML, or AML arising from MDS, or therapy-related AML), are shown to be better primary determinants of disease classification [8]. Two new categories have been suggested instead, irrespective of any history of MDS, namely AML with mutated *TP53* and AML with myelodysplasia-related gene mutations. The latter encompasses the prior entity of AML with mutated *RUNX1*, expanded to include the other eight genes shown to be related to AML arising from MDS or MDS/MPN as follows: *ASXL1*; *BCOR*; *EZH2*; *SF3B1*; *SRSF2*; *STAG2*; *U2AF1*; and *ZRSR2*. These mutations are strongly associated with a history of MDS or MDS/MPN and carry a poor prognosis [8].

In the absence of class-defining genetic abnormalities, a separate category of AML with myelodysplasia-related cytogenetic abnormalities incorporates cases with a complex karyotype (defined as ≥3 unrelated clonal chromosomal abnormalities) and/or other unbalanced chromosomal changes that previously fell into the prior AML-MRC category but lack a *TP53* mutation or myelodysplasia-related gene mutations, i.e., del(5q)/t(5q)/add(5q), –7/del(7q), +8, del(12p)/t(12p)/add(12p), i(17q), –17/add(17p)/del(17p), del(20q), or idic(X)(q13) [106].

The WHO has retained the 20% blast cut-off between MDS and AML for disease without AML-defining mutations because altering the threshold was felt to be arbitrary and carries the risk of overtreatment. Hence, the family of MDS with increased blasts (IBs) includes disease with <20% blasts, and this is further subclassified into IB1 (5–9% BM and/or 2–4% PB blasts), IB2 (10–19% BM and/or 5–19% BM blasts), and MDS with fibrosis (in which case 5–19% BM and/or 2–19% PB blasts are allowed) [15]. The AML-MRC category is maintained in the WHO guidelines, and it encompasses de novo AML or AML transformed from MDS or MDS/MPN. A key change is that diagnosis can no longer be made solely based on morphology, and it requires the identification of genetic abnormalities and/or history of MDS or MDS/MPN. The group of single-gene somatic mutations again includes the same group of myelodysplasia-associated genes mentioned above, present in the vast majority of AML arising from MDS or MDS/MPN with the exception of *RUNX1*, claiming insufficient unifying characteristics [15].

### 3.2. TP53-Mutated Disease

As described above, MDS and AML with *TP53* mutation are usually associated with a complex karyotype, carry a much worse prognosis and are recognized as separate disease entities by the ICC [9,102,108,109]. The *TP53* mutation is an independent predicting factor of poor response to cytarabine or hypomethylating agent (HMA)/Venetoclax-based therapy [110]. MDS with mutated *TP53* (i.e., blasts < 10%) requires the presence of a multi-hit *TP53* mutation or *TP53* mutations with variant allele frequency (VAF) > 10% and a complex karyotype, often with a loss of 17p. Cases with a complex karyotype but not a *TP53* mutation do not qualify for this category as they have different, more favorable outcomes [111]. Multi-hit *TP53* mutation is defined as two distinct *TP53* mutations (each with VAF > 10%), or a single *TP53* mutation with either 17p deletion (VAF > 50%) or loss of heterozygosity (LOH) in the 17p *TP53* locus [92]. In the absence of LOH information, a single *TP53* mutation in the context of a complex karyotype also qualifies for diagnosis. Monoallelic *TP53*-mutated MDS has different disease biology to multi-hit disease; thus, it is not included in this category and is instead categorized under MDS-NOS (if no EBs) or MDS with EB. On the contrary, monoallelic *TP53*-mutated MDS/AML or AML have a worse prognosis and are allowed in this category. TP53-mutated AML is not defined separately by the WHO [15].

### 3.3. Therapy-Related Disease

The therapy-related subgroup of MDS and AML is eliminated by the ICC, and therapy-related cases are now subclassified following primary diagnosis [102]. For example, MDS-NOS with single lineage dysplasia is therapy-related. This definition applies to therapy-related MDS, AML or MDS/AML, and AML progressed from MDS or MDS/MPN. Although it is recognized that secondary disease carries a worse prognosis, the priority is to first define the disease based on its morphologic and genetic features. The secondary myeloid neoplasm category has been preserved by the WHO and it includes therapy-related diseases (renamed to “myeloid neoplasm, post cytotoxic therapy or MN-pCT”), and diseases associated with germline predisposition (Down-syndrome-related AML falls into this category) [15]. Exposure to PARP inhibitors has been incorporated as a qualifying criterion for MN-pCT, whereas methotrexate exposure has been excluded, based on a lack of a significant association that has been reported [112,113]. Of note, AML transformed from another myeloid neoplasm is no longer classified as secondary by the WHO. AML arising from MDS or MDS/MPN is now classified under AML-MR (as per above) and AML arising from MPN is retained under the MPN category.

### 3.4. Other Changes Introduced by the ICC and the WHO

In the fourth edition of the WHO classification of myeloid neoplasms released in 2016, there was a significant overlap between the various AML subcategories [5]. A 20% blast cut-off was required for a diagnosis of AML, apart from cases of APL or AML with mutated CBF. Another major change introduced by both the ICC and the WHO groups is the expansion of AML-defining recurring genetic abnormalities to include t(9;11)(p21.3;q23.3)/*MLLT3::KMT2A* (or other *KMT2A* rearrangements), t(6;9)(p22.3;q34.1)/*DEK::NUP214*, inv(3)(q21.3q26.2)/*MECOM*(EVI1) (or other *MECOM* rearrangements) and *NPM1*, in-frame basic leucine zipper-region (bZIP) *CEBPA* mutations [114,115,116,117]. Biallelic *CEBPA* mutations are no longer required, with several studies showing favorable prognosis with monoallelic bZIP mutations [118,119]. In the case of *NMP1*-mutated AML, the WHO allows diagnosis irrespective of the blast percentage, highlighting the fact that *NMP1-*mutated neoplasms with blasts <20%, namely MDS and MDS/MPN cases, are associated with aggressive disease and rapid progression to overt AML within 12 months of diagnosis [120]. As newer, targeted therapies have emerged, early identification of the molecular subtype of the disease can lead to earlier initiation of treatment.

The classification algorithm of MDS and AML based on the new changes can be seen in Figure 1 and Figure 2.

## 4. Risk Stratification Changes in Myeloid Neoplasms

### 4.1. MDS

The IPSS-R has been used for years as the gold standard method for MDS risk stratification, as well as the numerous clinical implications this entails, such as clinical trial design and treatment modalities [121]. The IPSS-R is solely based on the hematologic and cytogenetic abnormalities of the disease and does not take into account individual mutations leading to clonality. As new knowledge around the significance of gene mutations in the prognostication of myeloid neoplasms has emerged, a new risk stratification scoring system has been recently suggested [30,31,73]. The IPSS-Molecular (IPSS-M) prognostic model was recently developed [122]. This new evidence is based on data from 2957 patients, from whom mutations of 152 different genes that were implicated in the development of myeloid neoplasms were assessed. The variables of the model were selected based on a stability selection algorithm that was applied to three different clinical outcomes as follows: overall survival (OS), leukemia-free survival (LFS) and AML transformation. As opposed to the IPSS-R, the absolute neutrophil count (ANC) is not considered part of the IPSS-M. At least one gene abnormality was found in 94% of the patients and the increasing number of mutations was negatively correlated with leukemia-free survival and thus disease severity [122].

This model uses hematologic (i.e., medullary blast percentage, hemoglobin, and platelet counts) and cytogenetic data (the five clusters that were included in the IPSS-R scoring system) and, in addition, includes molecular data, namely mutations identified in 16 “main effect” genes and 15 additional “residual” genes. Mutations in three of the main effect genes, *TP53 ^multihit^* (but not mono-allelic *TP53* mutations)*, MLL^PTD^* (partial tandem duplication) *and FLT3* (tyrosine kinase domain and internal tandem duplication) strongly correlated with adverse outcomes, highlighting the importance of screening for those genes at the time of diagnosis [122]. *FLT3* and *MLL^PTD^* mutations exhibited the strongest correlation with AML transformation [122]. Other main effect genes included *ASXL1, CBL, DNMT3A, ETV6, EZH2, IDH2, KRAS, NPM1, NRAS, RUNX1* and *SF3B1* (computational pattern *SF3B1^a^*) or *SF3B1^5q^, SRSF2,* and *U2AF* [122]. In the case of *SF3B1* mutations, there are three different clusters with different prognostic outcomes based on the co-mutation pattern identified. *SF3B1^5q^* (7% of *SF3B1* mutant cases) refers to the concomitant presence of isolated 5q deletion; *SF3B1^β^* (15% of the cases) refers to co-mutation of *SF3B1* along with any of the *BCOR, BCORL1, NRAS, RUNX1, SRSF2*, or *STAG2* genes; and *SF3B1^α^* (78% of the cases) is defined as any other mutant *SF3B1,* in which case 37% of the patients were found to have simple co-mutation patterns, including one of the *DNMT3A*, *TET2*, and/or *ASXL1* genes [122]. Favorable outcomes were observed for the *SF3B1^α^* group, whereas they were not for the other two groups [122]. Fifteen residual genes were also determined (*BCOR, BCORL1, CEBPA, ETNK1, GATA2, GNB1, IDH1, NF1, PHF6, PPM1D, PRPF8, PTPN11, SETBP1, STAG2* and *WT1*) based on adverse effects identified by univariate analysis and greater than 1% recurrence among all patients with MDS [122].

Six disease risk strata were identified based on this new model (instead of five per IPSS-R), which correlated with leukemia-free survival, overall survival and transformation to AML, as follows: very low (14%); low (33%); moderate low (11%); moderate-high (11%); high (14%); and very high (17%). Almost half (46%) of the patients of the cohort were re-classified based on IPSS-M. Of these, 74% were upstaged and 26% were downstaged [122]. The IPSS-M was applied to a separate Japanese cohort of 718 patients, and its discriminative power in terms of LFS was found to be superior to the IPSS-R across all key endpoints. The IPSS-M is applicable to both de novo disease and treatment-related MDS. It is a dynamic metric, i.e., it can be applied at any point during treatment. Its use in predicting treatment outcomes is yet to be determined.

### 4.2. AML

Both risk stratification and treatment management of AML are based primarily on the age, the molecular and cytogenetic footprint, and early measurable residual disease (MRD) status [123]. MRD positivity following intensive or non-intensive induction therapy is a major, independent, post-diagnosis poor prognostic factor [124,125,126].

The European Leukemia Net (ELN) has introduced some changes in regard to the genetic risk stratification of AML [127].

*FLT3*-ITD-mutated AML irrespective of the allelic ratio (formerly *FLT3*-ITD^high^ for biallelic mutation and *FLT3*-ITD^low^ for mono-allelic mutation) or the concurrent *NMP1* mutation status is now considered intermediate risk disease. This is based on evidence of disease modifying impact of midostaurin-based therapy [128].The previously recognized *RUNX1* and *ASXL1*-mutated AML categories are expanded to include the whole AML category with gene-related mutations classified under high-risk disease. This new category encompasses the nine above-mentioned gene mutations (*ASXL1, RUNX1, BCOR, EZH2, SF3B1, SRSF2, STAG2, U2AF1, ZRSR2*) associated with AML arising from MDS.*NMP1-*mutated AML with concurrent adverse-risk-associated cytogenetic abnormalities has been reclassified as a high-risk disease, based on new data that shows an association with worse outcomes [129].AML with in-frame monoallelic mutations in the basic leucine zipper region (bZIP) of *CEBPA* is now also classified under favorable-risk disease, along with biallelic mutated disease [130].The following cytogenetic abnormalities are added to the adverse risk group: t(3q26.2;v), involving the *MECOM* gene, and t(8;16)(p11.2;p13.3), associated with *KAT6A::CREBBP* gene fusion (39, 40). Last, hyperploidy with multiple trisomies or polysomies is no longer considered a complex karyotype and is therefore not classified as an adverse risk [14].

## 5. Current Management of Patients with High-Risk MDS and AML-MRC

Understanding the biology of MDS progression to AML-MRC is critical for the discovery of effective treatments and the development of a personalized therapeutic approach for patients with high-risk MDS and AML-MRC. Hypomethylating agents (HMAs) followed by allogeneic bone marrow transplantation (AlloBMT) remains the most frequently used approach [2]. Despite the requirement for further interpretation and exploitation of the new biological findings, significant progress has been made in the therapeutic approach of patients with MDS and AML-MRC. Venetoclax has now been approved in combination with hypomethylating agents by the United States Food and Drug Administration (FDA) [131], and targeted therapies, such as IDH1/2 inhibitors, have been effectively used in high-risk MDS and AML-MRC patients with IDH1/2 mutations [131]. Clinical trials testing novel agents, such as magrolimab and epretanoport, and anti-CD123 approaches in high-risk MDS and AML-MRC are currently ongoing [131]. Importantly, the new classifications of WHO and ICC have allowed clinicians to enroll high-risk MDS patients into AML clinical trials and AML-MRC patients into MDS clinical trials [15,132], which will undoubtedly expedite the results of the numerous ongoing clinical trials.

### 5.1. Current Standard Approaches

The current therapeutic approach for patients with MDS relies on (1) symptoms and clinical manifestation; (2) risk assessment using IPSS or Revised-IPSS or the most recently added IPSS-M calculator [122]; and (3) overall performance status and patient’s preference.

#### 5.1.1. AlloBMT

AlloBMT constitutes the only potentially curative approach for patients in the clinical spectrum of high-risk MDS and AML-MRC. The decision of AlloBMT candidacy is based on factors such as age, physical and mental status, and comorbidities and should be accessed individually [133]. Specifically, age is not an absolute exclusive criterion but a relative one. Although younger patients (<65 years old) tend to have longer overall survival, there is no significant difference in survival in older patients (>65 years old) [134].

The timing of the AlloBMT remains a significant challenge in clinical practice. Cutler et al. demonstrated that the optimal point of alloBMT is the time of progression to intermediate risk as per the IPSS-R. Particularly, low- and very-low-risk patients should be monitored closely and they should undergo delayed transplantation, but intermediate to very-high-risk patients benefit from early transplantation [135].

It is not yet clear which regimen is more effective at inducing remission in patients as a bridge to alloBMT. HMAs or induction chemotherapy have been used for this purpose [136] with an ongoing clinical trial comparing these two approaches before AlloBMT in patients with MDS (NCT01812252). The RICMAC and MDS200 trial failed to show any superiority of intensive chemotherapy over low-dose chemotherapy in patients with high-risk MDS or in AML-MRC patients [137]. Interestingly, another important consideration for MDS and AML-MRC patients undergoing alloBMT is the conditioning regimen. Scott et al. showed that even though a MAC regimen leads to increased transplant-related mortality, 4-year overall and relapse-free survival were significantly better compared to RIC, with AML-MRC patients deriving increased benefits compared to MDS patients [138].

Despite initial enthusiasm, maintenance strategies following AlloBMT for patients with high-risk MDS and AML-MRC have not led to significant survival benefits [139,140,141], except for specific subsets, such as *TP53*-mutated diseases [142].

#### 5.1.2. HMAs

The use of HMAs as first-line therapy is well established for patients with high-risk MDS and AML-MRC, and it is overall considered superior to intensive chemotherapy, partially due to a more tolerant and modest toxicity profile [143]. HMAs reduce the risk of death by 50% in high-risk MDS when used as first-line therapy in comparison with conventional care regimens, while most of the patients respond within 6 months of treatment [143]. Unfortunately, the HMA effect is transient, with responses usually maintained for less than 12 months, particularly in patients with adverse disease biology [144]. The prognosis of patients who progress to HMA is particularly dismal [144].

High-risk MDS and AML-MRC patients with poor response to HMAs showed a response rate of 41% and 32% when treated with chemotherapy and consequently underwent AlloBMT at a rate of 40% and 42%, respectively [145]. Despite the mechanisms of HMA resistance being the subject of intense pre-clinical research [101,146,147,148,149,150], novel combinational therapies based on these mechanisms have not yielded clinically significant results.

As opposed to post-AlloBMT maintenance, maintenance therapy with HMAs and particularly the oral formulation of azacitidine, CC-486 improve the outcomes of AML patients including a number of AML-MRC patients in first remission who did not receive AlloBMT [151].

#### 5.1.3. Venetoclax

Venetoclax (VEN, ABT-199) is a BH3 mimetic that inhibits the anti-apoptotic effects of BCL-2 proteins, inducing cell death in AML blasts, and was approved by the FDA in 2018 for the treatment of older patients with AML who are not eligible for intensive chemotherapy [152]. Azacitidine/venetoclax combination outperformed azacitidine alone in both de novo AML and AML-MRC based on overall survival (14.7 months vs. 9.6 months) and complete-remission rates (66.4% vs. 28.3%) [153]. AML with *IDH1/2* mutations demonstrated the highest sensitivity to the combination of azacitidine/venetoclax compared to azacitidine alone. Despite a more prolonged neutropenia and higher incidence of neutropenic fevers with combination therapy, the rate of discontinuation was similar between the two groups [153].

Recent evidence also supports that azacitidine/venetoclax is an effective combination in treatment-naïve high-risk MDS [154,155]. Liu et al. analyzed 1057 patients with high-risk MDS and AML-MRC treated with HMAs and venetoclax [156]. Fifty-six (56%) patients had complete remission and the overall response rate was 68% [156]. Subgroup analysis revealed that the complete-remission rate was 61% among MDS patients, 68% among patients with newly diagnosed AML-MRC and 39% for relapsed/refractory AML-MRC [156]. The combination of azacitidine/venetoclax is being evaluated in a phase 3 clinical trial for treatment-naïve high-risk MDS patients (NCT04401748).

The combination of azacitidine/venetoclax is also effective in relapsed/refractory MDS, as it outperformed azacitidine monotherapy and led to marrow response and complete-remission rates of 38.6% and 6.8%, respectively, within 1.5 months of treatment for approximately 9 months [157]. Of note, the response rate was stratified based on molecular profile rather than the conventional predicting tools (IPSS-R and blast percentage), with IDH2 and DNMT3A mutations showing the best response rate and TP53 mutations being associated with worse response rates.

Venetoclax-based regimens appear to be promising approaches as a bridge to alloBMT. It has been reported that high-risk MDS and AML-MRC patients responding to venetoclax-based therapy and then receiving AlloBMT have a 1-year overall survival rate of 75–79% with a relapse rate of 16.7% within the first 12–14 months [158,159].

It remains unclear if azacitidine/venetoclax is a better approach compared to induction chemotherapy for fit patients with high-risk MDS and AML-MRC. Ongoing clinical trials focus on comparing induction chemotherapy to the azacitidine/venetoclax combination, with emphasis on the outcomes of patients with AML-MRC (NCT04801797).

#### 5.1.4. Chemotherapy

Low-dose cytarabine and induction chemotherapy have been evaluated for the management of patients with AML-MRC with responses of approximately 50–60% but a median survival of 6.5 months [160]. Prior treatment with HMAs or lenalidomide and longer time to transform to AML-MRC were associated with worse response [160]. A recent retrospective analysis showed high response rates of induction chemotherapy, followed by AlloBMT in MDS and MDS/MPN patients with *NPM1* mutations [116], suggesting that molecular profile may be critical for the identification of MDS/AML-MRC patients that may benefit from induction therapy. Lancet et al. compared the activity of CPX-351, a dual-drug liposomal encapsulation of cytarabine and daunorubicin that delivers a synergistic 5:1 drug ratio into leukemia cells to a greater extent than normal bone marrow cells, to the activity of 7 + 3 in older individuals with AML-MRC [161]. CPX-351 demonstrated significantly improved remission rates and overall survival, despite a prolonged time to neutrophil and platelet engraftment [161]. This survival benefit of CPX-351 over 7 + 3 was maintained after 5 years of follow-up based on a subsequent study [162]. Overall, these results indicate that chemotherapy may be an option for fit individuals with high-risk MDS with elevated blasts or AML-MRC.

### 5.2. Specific Molecular Subtypes

As described above, the molecular profile of MDS and AML-MRC define the disease biology and affect its natural history independent of the disease stage at diagnosis. Consistently, treatment approaches for specific molecular subtypes are currently the focus of intense research.

#### 5.2.1. *TP53* Mutated High-Risk MDS and AML-MRC

Biallelic *TP53* alteration is one of the most established adverse genomic subtypes of high-risk MDS and AML-MRC and, as described above is associated with particularly poor outcomes. HMAs and HMAs/venetoclax combinations show some efficacy in *TP53*-mutated high-risk MDS and AML-MCR [153,163], but the incidence of relapse and long-term overall survival remains disappointing [164]. Similarly, it remains unclear if chemotherapy provides any significant benefit for *TP53*-mutated high-risk MDS and AML-MRC patients [91].

AlloBMT is the only approach that can prolong the survival of *TP53*-mutated MDS and AML-MRC patients [110]. However, *TP53*-mutated patients, compared with *TP53* wild-type patients, showed a higher relapse rate after alloBMT [34]. The achievement of a lower burden of TP53 mutation before alloBMT has been associated with improved survival [34].

These results clearly support that TP53-mutated MDS and AML-MRC patients represent an unmet need. Recent studies have led to the introduction of two novel agents with promising efficacy for these patients.

Eprenetapopt (ARP-246) is a small molecule that stabilizes when converted to methylene quinuclidinone, the misfolded P53 core domain, and restores the function of the mutated protein [165,166,167]. Pre-clinical studies supported that APR-246 has synergistic effects when combined with azacitidine, ara-c and daunorubicin in *TP53*-mutated AML cells [168,169]. Despite the encouraging findings of two phase I/II trials showing that the APR-246/azacitidine combination resulted in high response and complete-remission rates, as well as high rates of *TP53*-mutation clearance [170,171], the survival benefit of this combination over azacitidine alone for *TP53*-mutated high-risk MDS/AML-MRC has not been confirmed. However, recent data support that the APR-246/azacitidine combination has promising efficacy as a maintenance therapy following AlloBMT for *TP53*-mutated MDS and AML patients [142]. Finally, evaluation of the triplet combination azacitidine/venetoclax/APR-246 is ongoing in high-risk MDS and AML-MRC with adverse cytogenetic and molecular features [172].

Magrolimab is a monoclonal antibody that targets the CD47 surface marker of tumor cells, also known as the “don’t eat me” signal, promoting their phagocytosis by macrophages [173]. Among *TP53*-mutated MDS and AML-MRC patients, the combination of azacitidine/magrolimab showed a response rate of about 40%, which is similar to *TP53*-unmutated patients with anemia due to hemolysis being a well-described side effect [174]. Of note, the combination of magrolimab/azacitidine/venetoclax demonstrated complete-remission rates higher than 80% in newly diagnosed AML patients with ELN adverse risk, suggesting that this approach may become the new standard of care for high-risk MDS/AML-MRC patients who are ineligible for induction chemotherapy but can tolerate this triple therapy [175]. Magrolimab-based therapies are currently being compared to standard of care, such as azacitidine and azacitidine/venetoclax, in various ongoing phase three clinical trials (NCT05079230, NCT04313881, NCT04435691, NCT04778397, NCT04435691).

#### 5.2.2. *IDH1/2* Mutated High-Risk MDS and AML-MRC

*IDH1/2* mutations are relatively uncommon in the chronic phase of MDS, but their incidence increases with the transformation to AML-MRC and are associated with overall poor outcomes [176,177]. The IDH1 inhibitor ivosidenib results in high complete-remission rates in patients with *IDH1*-mutated relapsed/refractory AML [178] and newly diagnosed AML [179]. Phase III clinical trials have shown that the addition of ivosidenib to azacitidine significantly increases remission rates and prolongs the survival of patients with *IDH1*-mutated AML, including AML-MRC [180]. This led to the FDA approval of this combination for patients with *IDH1*-mutated AML who are ineligible for induction chemotherapy. Enasidenib is an IDH2 inhibitor that leads to response rates of 30–50% in relapsed/refractory *IDH2*-mutated AML, including AML-MRC [181,182]. Enasidenib/azacitidine was evaluated for high-risk *IDH2*-mutated MDS, demonstrating a response rate of 74% and a complete remission of 26% in 27% of patients proceeding to alloBMT and survival after 2 years [182]. Finally, enasidenib has been evaluated as maintenance therapy for high-risk MDS and AML-MRC patients with *IDH2* mutation with overall, progression-free and relapse rates being 74%, 69% and 16%, respectively, and relatively rare severe adverse effects [183]. Ongoing clinical trials evaluate the efficacy of IDH1/2 inhibitors in MDS and AML-MRC patients (NCT03503409, NCT04603001, NCT04493164, NCT03471260) and their combination with venetoclax, azacitidine and intensive chemotherapy (NCT03471260, NCT03839771).

## 6. Conclusions

High-risk MDS is a clonal myeloid neoplasm characterized by genetic abnormalities in stem and progenitor hematopoietic cells with a high incidence of disease progression to a disease stage with a high percentage of blasts defined as AML-MRC or AML with myelodysplasia-related gene mutations or MDS/AML. Recent pre-clinical data support that genetic alterations, such as chromosomal abnormalities and somatic mutations, are detectable in hematopoietic cells even at the earlier stages of the disease and predict disease progression defining its natural history and biology. These results suggest that MDS and AML-MRC represent a continuum of the same disease and suggest that early-risk stratification and intervention may be a particularly promising approach for patients with MDS and a high risk of progression. These advances are reflected in the new changes in the classification of MDS and AML and are utilized in the evaluation of novel therapies.

## Figures and Tables

**Figure 1 ijms-24-05018-f001:**
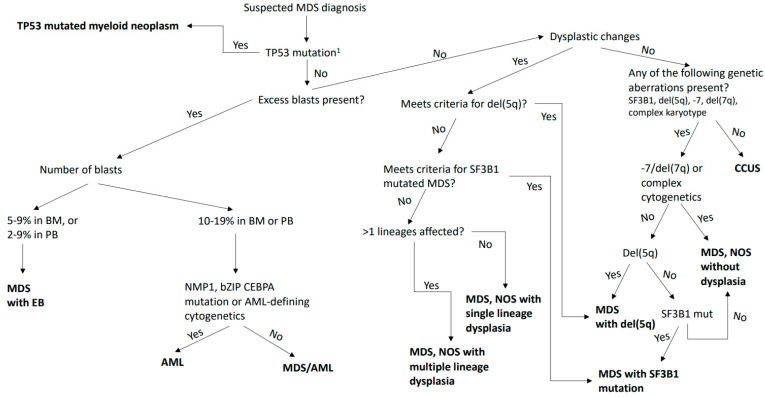
The classification algorithm of MDS based on the new changes. ^1^ Diagnosis of MDS/AML or AML with mutated *TP53* can be made with any somatic *TP53* mutation with VAF > 10%, whereas MDS with mutated *TP53* requires the presence of a multi-hit *TP53* mutation or a *TP53* mutation (VAF > 10%) and a complex karyotype, often with loss of 17p if LOH information is not available. A multi-hit *TP53* mutation is defined as either two distinct *TP53* mutations, each with VAF > 10%, or a single *TP53* mutation with either 17p deletion on cytogenetics, VAF ≥ 50%, or copy-neutral LOH at the 17p locus. Abbreviations: MDS, myelodysplastic syndromes; AML, acute myeloid leukemia; EBs, excess blasts; NOS, not otherwise specified; CCUS, clonal cytopenia of undetermined significance.

**Figure 2 ijms-24-05018-f002:**
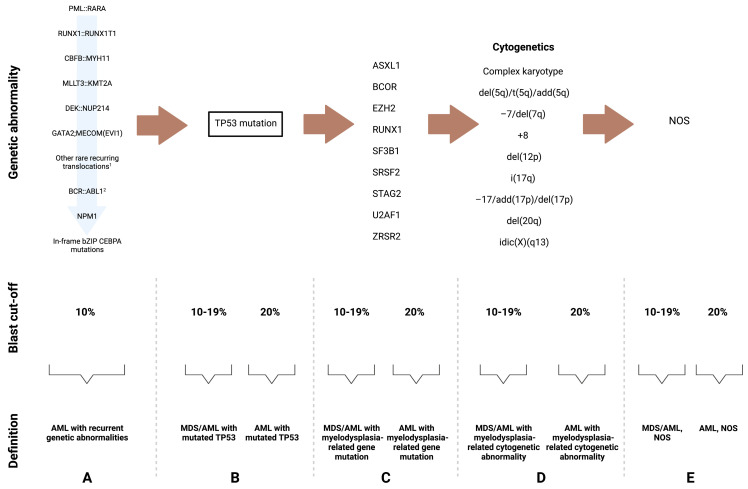
The classification algorithm of AML based on the new changes. (**A**) AML with recurrent genetic abnormalities takes precedence in the diagnostic algorithm, and the blast cut-off for diagnosis is lowered to 10%, with the exception of BCR::ABL-mutated AML. Within this group, PML::RARA mutations supersede RUNX1::RUNX1T1 mutations, followed by CBFB:MYH11, then MLLT3:KMT2A, then DEK::NUP214, then GATA2;MECOM(EV11), followed by other rare recurring translocations as depicted below, then BCR.:ABL1, then NPM1, and last in-frame bZIP CEBPA mutations. Among the remaining categories, (**B**) TP53-mutated AML supersedes (**C**) AML with myelodysplasia-related gene mutations and the latter supersedes (**D**) AML with myelodysplasia-related cytogenetic abnormalities. If none of the aforementioned genetic abnormalities are present, then AML is defined as (**E**) not-otherwise specified. ^1^ Acute myeloid leukemia (AML) with other rare recurring translocations includes the following: t(1;3)(p36.3;q21.3)/*PRDM16::RPN1*, t(3;5)(q25.3;q35.1)/*NPM1::MLF1*, t(8;16)(p11.2;p13.3)/*KAT6A::CREBBP*, t(1;22)(p13.3;q13.1)/*RBM15::MRTF1*, t(5;11)(q35.2;p15.4/*NUP98::NSD1*, t(11;12)(p15.4;p13.3)/*NUP98::KMD5A*, *NUP98* and other partners, t(7;12)(q36.3;p13.2)/*ETV6::MNX1*, t(10;11)(p12.3;q14.2)/*PICALM::MLLT10*, t(16;21)(p11.2;q22.2)/*FUS::ERG*, t(16;21)(q24.3;q22.1)/*RUNX1::CBFA2T3*, inv(16)(p13.3q24.3)/*CBFA2T3::GLIS2.*
^2^ The MDS/AML category is not applicable due to its overlap with progression of *BCR::ABL1*-positive CML. Abbreviations: MDS, myelodysplastic syndromes; AML, acute myeloid leukemia; NOS, not otherwise specified.

**Table 1 ijms-24-05018-t001:** The incidence, biologic mechanisms and clinical significance of genetic abnormalities in MDS and AML-MRC.

Genetic Alteration	Incidence in MDS/AML	Possible Biologic Mechanism	Clinical Significance	References
Chromosomal abnormalities				
Deletion of 5q	10–15%	Defective ribosomal biogenesis leads to degradation of MDM2	Overall good prognosis	[16,17,18,19,20,21]
Deletion of chromosome 7/7q	~20%	Loss of chromosome 7 is linked with compromised expression of interferon gamma pathway genes.	Overall poor prognosis	[23,24,25,26,27,28]
Complex karyotype (CK)	30%	Mutations in *TP53* and frequent LOH of this gene	Poor prognosis/survival	[29,30,31,32,33,34,35]
Somatic mutations				
*SF3B1*	10–15%	Alterations in the spliceosome that frequently lead to non-sense-mediated decay (NMD)	Overall good prognosis but specific variants may have different impact	[39,40,41,42]
*SRSF2*	10–15%	Alteration of splicing and DNA damage accumulation.	Poor prognosis	[43,44,45,46]
*U2AF1*	5–15%	Suppression of ATG7 levels is associated with increased oxidative stress and chromosomal instability	Poor prognosis/survival	[47,48,49]
*ZRSR2*	5%	Retention of U12-dependent introns causing aberrant splicing	Negative impact on survival less strong compared to *SRSF2* and *U2AF1*	[31,50,51]
*STAG2*	~10%	Dysplasia caused by abnormal hematopoietic stem cell maturation along with additional oncogenic mutations	Poor prognosis	[53,54,55,56,57]
*EZH2*	4–5%	Gain of function mutations lead to improper control of cell proliferation	Poor prognosis/survival	[58,59,60,63,65,93]
*BCOR*	<5%	Loss of function mutations causing increased myeloid cell proliferation	Neutral impact on survival	[69,70,71,72]
*RUNX1*	10–15%	Mutated gene transcripts block gene activation responsible proper hematopoiesis	Poor prognosis/survival	[67,74,76,77,78,79]
*NRAS*/*KRAS*	4–5%	*RAS* gene mutations activate RAS/Raf/MEK and PI3K/AKT pathways	Poor prognosis/survival	[80,81,82,83,84,85]
*FLT3-ITD*	0.6–6%	Mutations cause constitutive auto-phosphorylation of the FLT3 receptor leading to activation of downstream pathways	Poor prognosis/survival	[11,88,89]
*TP53*	5–10%	Mutations lead to impaired DNA damage repairs, LOH is critical for the prognosis	Poor prognosis/survival	[90,91,92]

**Table 2 ijms-24-05018-t002:** Pre-malignant myeloid clonal conditions.

	Description	Clonality	Cytopenia	Dysplasia	Increased Blasts	Genetic Testing
CH	Lineage derived from a single clone	Yes	No	No	No	General term to define somatic mutations or cytogenetic abnormalities or copy number mutations ^1^
CHIP	Lineage derived from a single clone	Yes	No	No	No	Somatic mutation in a myeloid neoplasm driver gene with VAF ≥ 2%, or Non-MDS defining clonal cytogenetic abnormality in the absence of a myeloid neoplasm or cytopenia
CCUS	Lineage derived from a single clone + Cytopenia	Yes	Yes	No	No	Somatic mutation in a myeloid neoplasm driver gene with VAF ≥ 2% ^2^
VEXAS syndrome	CH + anemia	Yes	Yes	No	No	Somatic mutation in *UBA1* gene

^1^ Can be seen post-treatment of myeloid neoplasms or even solid tumors, in which case the clinical implications might differ. ^2^ Higher VAF is associated with higher progression to MDS [107]. Abbreviations: CH, clonal hematopoiesis; CHIP, clonal hematopoiesis of indeterminate potential; CCUS, clonal cytopenia of undetermined significance; VEXAS syndrome (vacuoles, E1 enzyme, X-linked, autoinflammatory, somatic).

## Data Availability

Not applicable.

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
