# Peer review of "Understanding the Continuum between High-Risk Myelodysplastic Syndrome and Acute Myeloid Leukemia"

_ijms, 2023, doi:10.3390/ijms24055018_

Round 1

Reviewer 1 Report

Zavras and Sinanidis et al. described the summary of high-risk myelodysplastic syndrome (MDS) and acute myeloid leukemia (AML). Basic information including molecular aspects of progression from MDS to AML are well explained. Also, the authors are considering the 5th edition of the new World health organization (WHO) classification. This review is very interesting and informative for readers.

Only minor things; 

1. Abbreviations of WHO and ICC are duplicated in lines 271 and 272.

2. In lines 397 and 407, the extra word “1” is observed. Also, the extra “Table” is in line 409.

3. Extra space is observed in Table 1 under the "Somatic mutation"

Author Response

  1. Abbreviations of WHO and ICC are duplicated in lines 271 and 272.

We thank the reviewer for pointing this out. We have corrected this point in our revised manuscript. 

  1. In lines 397 and 407, the extra word “1” is observed. Also, the extra “Table” is in line 409.

The superscript numbers “1” and “2” under figure 2 correspond to the numbers on the figure, so these were maintained. The word “table” was a typo and was removed. 

  1. Extra space is observed in Table 1 under the "Somatic mutation"

We have corrected this point in our revised manuscript. 

Reviewer 2 Report

Zavras et.al. reviewed the latest detail of classification in MDS which is currently distinguished as mixed pre-myeloid malignancies. The article would be worth to read who demand to understand MDS classification and each risk mutation for AML progression thought these are some improve points wrote down below. 

The author wrote down “SRSF2, SF3B1, ZRSF2 and U2AF1, 129 EZH2, BCOR and STAG2 are >95% specific for AML-MRC” (line129) how do you describe ZRSF2 and BCOR of current preclinical knowledge and add for Table 1? 

The article mainly focused on “current pre-clinical studies supporting that high-risk MDS and AML-MRC represent a continuum and provide a novel understanding of the mechanisms of disease progression”  and “on the most recent changes in the classification of MDS and AML-MRC and the implications of these advances in the management of patients with these myeloid neoplasms” (Line65-69), how do you change the abstract and paragraph due to that:

“3. SINGLE-CELL ANALYSIS REVEALS BIOLOGIC MECHANISMS IMPLICATED IN MDS PROGRESSION TO AML-MRC (line235)” should be under “2. GENETIC ALTERATIONS UNDERLIE MDS PROGRESSION TO AML-MRC“, as 2.3

“5. MDS and AML with myelodysplasia-related gene mutations as a disease continuum” (line276) should be under  “4. CHANGES IN THE DEFINITION AND CLASSIFICATION OF MDS AND AML-MRC” as 4.1

5.1. TP53-mutated disease (line 342) should be under “4. CHANGES IN THE DEFINITION AND CLASSIFICATION OF MDS AND AML-MRC” 

5.2. Therapy-related disease should be under “4. CHANGES IN THE DEFINITION AND CLASSIFICATION OF MDS AND AML-MRC” 

  6. Other changes introduced by the ICC and the WHO  should be under “4. CHANGES IN THE DEFINITION AND CLASSIFICATION OF MDS AND AML-MRC” 

7. Risk Stratification Changes in Myeloid Neoplasms, 8. CURRENT MANAGEMENT OF PATIENTS WITH HIGH-RISK MDS AND AML-MRC, 9. Current standard approaches and 10. Specific molecular sub-types are difficult to follow. These should be gathered and focus on current and new approaches for therapy by each MDS/AML mutation.

I’m confusing the words of the difference between AML-MRC and MDS/AML. If the classification wrote down as MDS/AML, how do you use only MSD/AML?

Line 106 chromosome 9 should be chromosome 7 

Line 382 the sort of sentence should be wrong

Figure 2 is difficult to follow because the pages are separated, should include a page

Author Response

1)         The author wrote down “SRSF2, SF3B1, ZRSF2 and U2AF1, 129 EZH2, BCOR and STAG2 are >95% specific for AML-MRC” (line129) how do you describe ZRSF2 and BCOR of current preclinical knowledge and add for Table 1? 

We thank the reviewer for his comment. In our revised manuscript we included sections summarizing the literature on the biology of BCOR and ZRSR2 mutations and their impact on the outcomes of MDS patients (lines 162-165, 191-199). We also included this information to our Table 1.

2)         The article mainly focused on “current pre-clinical studies supporting that high-risk MDS and AML-MRC represent a continuum and provide a novel understanding of the mechanisms of disease progression”  and “on the most recent changes in the classification of MDS and AML-MRC and the implications of these advances in the management of patients with these myeloid neoplasms” (Line65-69), how do you change the abstract and paragraph due to that:

We agree with the reviewer on these points. In the revised manuscript, the abstract has been updated with the addition of the important findings from the pre-clinical studies mentioned in the text. 

We believe that the recent classification changes by the ICC and the WHO as well as the advances in terms of the of management of these patients have been adequately described in the abstract, thus we believe no further changes are necessary.

3)         “3. SINGLE-CELL ANALYSIS REVEALS BIOLOGIC MECHANISMS IMPLICATED IN MDS PROGRESSION TO AML-MRC (line235)” should be under “2. GENETIC ALTERATIONS UNDERLIE MDS PROGRESSION TO AML-MRC“, as 2.3

We have revised this in our manuscript.

4)         “5. MDS and AML with myelodysplasia-related gene mutations as a disease continuum” (line276) should be under  “4. CHANGES IN THE DEFINITION AND CLASSIFICATION OF MDS AND AML-MRC” as 4.1

This point has been revised. It is now showing as “3.1” since the original paragraph 3 was incorporated to paragraph 2, as indicated above. 

5)         5.1. TP53-mutated disease (line 342) should be under “4. CHANGES IN THE DEFINITION AND CLASSIFICATION OF MDS AND AML-MRC” 

We have revised this in our manuscript.

6)         5.2. Therapy-related disease should be under “4. CHANGES IN THE DEFINITION AND CLASSIFICATION OF MDS AND AML-MRC” 

We have revised this in our manuscript.

7)         6. Other changes introduced by the ICC and the WHO  should be under “4. CHANGES IN THE DEFINITION AND CLASSIFICATION OF MDS AND AML-MRC” 

We have revised this in our manuscript.

8)         7. Risk Stratification Changes in Myeloid Neoplasms, 8. CURRENT MANAGEMENT OF PATIENTS WITH HIGH-RISK MDS AND AML-MRC, 9. Current standard approaches and 10. Specific molecular sub-types are difficult to follow. These should be gathered and focus on current and new approaches for therapy by each MDS/AML mutation.

We thank the reviewer for this comment. In the revised manuscript we performed extensive revision of the section 8. CURRENT MANAGEMENT OF PATIENTS WITH HIGH-RISK MDS AND AML-MRC to make it easier to follow by making it more focused and concise. It is challenging to create multiple sections based on molecular prolife as TP53 and IDH1/2 mutated MDS/AML-MRC are the only subtypes with approved therapies. Lines 505-714.

9)         I’m confusing the words of the difference between AML-MRC and MDS/AML. If the classification wrote down as MDS/AML, how do you use only MSD/AML?

We appreciate the reviewer’s time and effort to improve our review article. MDS/AML is a more recent term used by the ICC to indicate what was previously known as MDS with increased blasts, between 10-19% (lines 321-324). AML-MRC is no longer used by the ICC, however is still used by the WHO to refer to AML (with blast cut-off ≥20%) de novo AML or AML arising from MDS or MDS/MPN, and it requires identification of genetic abnormalities and/or history of MDS or MDS/MPN (lines 344-348). 

Throughout the paper, the term AML-MRC is used to describe overt AML with MDS-related genetic abnormalities. 

10)       Line 106 chromosome 9 should be chromosome 7 

We have corrected this in our revised manusctipt.

11)       Line 382 the sort of sentence should be wrong

Another major change introduced by both the ICC and the WHO groups includes the expansion of AML-defining recurring genetic abnormalities, in which the blast cut-off is lowered to 10%, including t(9;11)(p21.3;q23.3)/MLLT3::KMT2A (or other KMT2A rearrangements), t(6;9)(p22.3;q34.1)/DEK::NUP214, inv(3)(q21.3q26.2)/MECOM(EVI1) (or other MECOM rearrangements), NPM1, in-frame basic leucine zipper region (bZIP) CEBPA mutations

We have revised this sentence in our revised manuscript.

Reviewer 3 Report

This review paper provided by Zavras et al., provided comprehensive summary of the etiology and advancement of pre-clinical information about MDS.  The authors also provided an organized scheme (Fig.1) for better diagnostic of the disease and its progression.   Useful information was pulled together for a complete overview of MDS.

Author Response

We thank the reviewer for his comments and the time dedicated to our manuscript.

Reviewer 4 Report

This manuscript is a comprehensive review of high-risk MDS and AML in terms of genetic abnormalities. The authors summarize the current pre-clinical studies supporting that high-risk MDS and AML-MRC represent a continuum and provide a novel understanding of the mechanisms of disease progression. This manuscript was well written and provides useful information for hematologists.

I have no further comments.

Author Response

(The authors gave the same response as above.)

Round 2

Reviewer 2 Report

I think it improve and easier to read by revise.

Some minor points below,

1. In my checking PDF, I couldn’t see Figure 2. Please care about the missing pages

2. Re-numbered paragraph number would be wrong (for example, Line 438 7 would be 4).  

Author Response

Reviewer 2

I think it improve and easier to read by revise.

We thank the reviewer for his comment.

Some minor points below,

1. In my checking PDF, I couldn’t see Figure 2. Please care about the missing pages

We have added the figure 2 in the main text.

  1. Re-numbered paragraph number would be wrong (for example, Line 438 7 would be 4).  

We agree with this point and we have revised the paragraph numbers accordingly.